# Dual Role of DUOX1-Derived Reactive Oxygen Species in Melanoma

**DOI:** 10.3390/antiox12030708

**Published:** 2023-03-13

**Authors:** Irene Pardo-Sánchez, Sofía Ibañez-Molero, Diana García-Moreno, Victoriano Mulero

**Affiliations:** 1Departamento de Biología Celular e Histología, Facultad de Biología, Universidad de Murcia, 30100 Murcia, Spain; 2Instituto Murciano de Investigación Biosanitaria (IMIB)-Pascual Parrilla, 30120 Murcia, Spain; 3Centro de Investigación Biomédica en Red de Enfermedades Raras (CIBERER), Instituto de Salud Carlos III, 28029 Madrid, Spain

**Keywords:** melanoma, DUOX1, oxidative stress, reactive oxygen species, metastasis, zebrafish

## Abstract

Melanoma is the most serious type of skin cancer. Inflammation and oxidative stress play an essential role in the development of several types of cancer, including melanoma. Although oxidative stress promotes tumor growth, once cells escape from the primary tumor, they are subjected to a more hostile environment, with higher levels of oxidative stress typically killing most cancer cells. As Dual Oxidase 1 (DUOX1) is a major producer of reactive oxygen species (ROS) in epithelia, we used allotransplantation and autochthonous melanoma models in zebrafish together with in silico analysis of the occurrence and relevance of *DUOX1* expression of the skin cutaneous melanoma (SKCM) cohort of The Cancer Genome Atlas (TCGA) to address the role of this enzyme in the aggressiveness of melanoma cells in vivo. It was found that high transcript levels of the gene encoding DUOX1 were associated with the poor prognosis of patients in the early-stage melanoma of TCGA cohort. However, *DUOX1* transcript levels were not found to be associated to the prognosis of late-stage SKCM patients. In addition, the transcript level of *DUOX1* in metastatic SKCM was lower than in primary SKCM. Using zebrafish primary melanoma and allotransplantation models, we interrogated the role of DUOX1 in vivo. Our results confirmed a dual role of DUOX1, which restrains melanoma proliferation but promotes metastasis. As this effect is only observed in immunocompromised individuals, the immune system appears to be able to counteract this elevated metastatic potential of DUOX1-deficient melanomas.

## 1. Introduction

Skin cutaneous melanoma (SKCM) is the fifth most common cancer and the most serious skin cancer [1]. The incidence of malignant SKCM has been increasing worldwide, from a rare skin cancer a century ago to an estimate in 2021 of more than 100,000 new cases in the United States and around 7000 deaths caused by this disease [2,3]. SKCM is now considered a multifactorial disease resulting from genetic susceptibility and environmental exposure [4]. It is caused by the malignant transformation of melanocytes, a type of skin cell that produces melanin to protect against exposure to ultraviolet (UV) light [5]. The main environmental risk factor for developing malignant SKCM is precisely UV exposure [6,7]. The skin is constantly exposed to UV light which can induce mutagenesis. After exposure to UV radiation, melanin can produce superoxide radicals that cause lethal cellular injury, but melanin is also important for skin homeostasis by absorbing most UV radiation [8].

SKCM begins as a proliferation of normal melanocytes to form a nevus, in which the melanocytes remain confined to the epidermis. Subsequently, the nevus may acquire atypical growth, and a dysplastic nevus develops. Unlimited radial growth then begins, followed by a vertical growth phase, which crosses the basement membrane to form a tumor and finally successfully spreads throughout the body in the form of metastatic tumors [9,10]. The Clark level system classifies SKCM based on how many layers of the skin it has grown into. Level 1: it is confined to the epidermis; Level 2: it has invaded the papillary dermis; Level 3: it has invaded throughout the papillary dermis and is touching on the reticular dermis; Level 4: it has invaded the reticular dermis; and Level 5: it has invaded the fat under the dermis [11].

It has been proposed that SKCM can be divided into four subtypes according to mutation in the three most prevalent mutated genes: BRAF (B-RAF proto-oncogene, serine/threonine kinase), NRAS (NRAS proto-oncogene, GTPase), NF1 (neurofibromin 1) and Triple WT (wild type), without mutations in any of these genes [12]. SKCMs often harbor BRAF mutations (≈50%) and, to a lesser degree, NRAS mutations (28%). BRAF is the main oncogene found in both malignant melanoma and in benign nevi. BRAF-V600E mutations are present in most nevus, so other lesions would have to occur to develop melanoma. However, this fact points to the important role of this oncogene in melanocyte transformation at early stages. The RAS-RAF-MEK-ERK (mitogen-activated protein kinase, MAPK) and the PTEN-PI3K-AKT (AKT) signaling pathways are constitutively activated in SKCM through multiple mechanisms and play several key roles in its development and progression [13]. Activation of the MAPK pathway culminates in the regulation of gene transcription in the nucleus by the extracellular signal-regulated kinase ERK, which phosphorylates several cellular substrates enabling proliferation [14]. In melanocytes, BRAF induces the activation of MEK kinase, which activates ERK, the end effector of the MAPK cascade, via phosphorylation, resulting in continuous stimulation of cell proliferation and tumor growth [15]. The second major pathway of cell growth regulation is the signal transduction PTEN/PI3K/AKT cascade, depending also on RAS [16]. It has been demonstrated that the activation of AKT results in the development of more metastatic SKCM in mice [17]. In addition, microphthalmia-associated transcription factor (MITF) is involved in the control of proliferation and differentiation of melanocytes and is also associated with SKCM development and progression [18]. The role of MITF in SKCM is very complex: although melanoma cells expressing MITF at a high level can either differentiate or proliferate, low activity of MITF is related to stem cell-like or invasive potential [19].

SKCM is one of the most immunogenic types of cancer [20]. The immunogenicity of a tumor is the ability to induce adaptive immune responses that can prevent tumor growth [21]. In addition, many studies support the concept that innate immunity plays a crucial role in the development, growth, and prognosis of malignant melanoma [22]. Furthermore, oxidative stress also plays a key and complex role in melanoma: melanoma cells in the blood experienced oxidative stress that was not observed in subcutaneous tumors, and thus oxidative stress limits distant metastasis of melanoma cells in vivo, raising the possibility that treatment with antioxidants may favor the progression of this cancer by promoting metastasis [23]. It has also been shown that immune cells, such as neutrophils and macrophages, are attracted to transformed cells at surprisingly early stages. An important attractant molecule is H_2_O_2_ [24], which is also an essential early damage signal responsible for driving neutrophils to wounds [25,26,27]. H_2_O_2_, which is produced by both transformed cells and their healthy neighbors, can diffuse away from its site of generation, and may act as a signaling factor. Furthermore, it is becoming clear that H_2_O_2_ plays a fundamental role in cell proliferation, migration, metabolism, and cell death [28].

Dual oxidase 1 (DUOX1) belongs to the NADPH oxidase (NOX) family of transmembrane enzymes that transfer electrons across biological membranes, reducing O_2_ to O_2_^−^ or H_2_O_2_ [29]. The NOX family is formed of seven members, which are NOX1 to NOX5 and DUOX1 and 2. All NOX isoforms have six highly conserved transmembrane domains, one NADPH binding site in the C-terminal region, one FAD binding site, and two histidine-linked heme groups in the transmembrane domains III and IV. NOX5 and DUOX 1 and 2 have an intracellular calcium-binding site that is involved in their activation [29]. Although DUOX1 was initially described as a source of H_2_O_2_ in the thyroid, new data suggests that DUOX1 is a major source of H_2_O_2_ in several epithelia, including skin [30]. Furthermore, the relevance of this enzyme in cancer initiation, progression and responses to therapy has recently been reported. For example, inhibition of DUOX1 impairs the recruitment of neutrophils and macrophages to oncogenic-transformed cells in zebrafish, resulting in reduced growth of the transformed cell clones [24]. Similarly, genetic inhibition of DUOX1 in human lung carcinoma cells promotes epithelial-to-mesenchymal transition, which is involved in metastasis, resistance to tyrosine kinase inhibitors, and invasiveness [31]. In addition, either inhibition of DUOX1 or treatment of cells with catalase to scavenge H_2_O_2_ abrogate ionizing radiation-induced DNA damage in human thyroid cells [32].

The zebrafish has been increasingly used in cancer research due to its unique advantages, including its easy genetic and pharmacological manipulation and the possibility of in vivo tracking of cancer cells using available transgenic lines with fluorescent reporters [33]. This model is specifically useful for melanoma research since early melanoma transformation can be visualized using the *crestin:GFP* transgenic line, which expresses green fluorescent protein (GFP) driven by the neural crest-specific promoter *crestin* [34]. In addition, a recently developed method, named MiniCoopR, allows the screening of large numbers of candidate melanoma modifiers [35]. This system combines the expression of the *mitfa* (melanocyte inducing transcription factor a) minigene and the candidate melanoma modifier gene driven by the *mitfa* promoter to express candidate genes in rescued melanocytes of zebrafish with loss-of-function alleles of *mitfa*. In the present study, we used these zebrafish models of melanoma to address the role of DUOX1 in the aggressiveness of melanoma cells in vivo. Our results confirmed a dual role of DUOX1, which restrains melanoma proliferation but promotes metastasis. As this effect is only observed in immunocompromised individuals, the immune system appears to be able to counteract this elevated metastatic potential of DUOX1-deficient melanomas. The relevance of this enzyme in melanoma was further confirmed by our analysis of the SKCM patient cohort of TCGA that showed that high transcript levels of *DUOX1* were associated with poor survival of early-stage but not of late-stage SKCM patients. In addition, the transcript level of *DUOX1* in metastatic SKCM was lower than in primary SKCM. Collectively, these results highlight the complex crosstalk between oxidative stress and immunity in cancer.

## 2. Methods and Materials

### 2.1. Human SKCM Dataset Analysis

Normalized gene expression and patient survival data were downloaded on 9 January 2023 from the SKCM repository of TCGA (cohort Firehose Legacy, https://datacatalog.mskcc.org/dataset/10490) from cBioPortal database (https://www.cbioportal.org/). This cohort includes 480 patient samples. No cleaning filter was applied, and all available patient data for each study were used: 462 samples for expression analysis and 316 samples for survival analysis. Clark levels at diagnosis were used for patient stratification: I, II and III stages were categorized as early-stage SKCM, while IV and V as late-stage SKCM. Gene expression plots and regression curves for correlation studies were obtained using GraphPad Prism 5.03 (GraphPad Software).

### 2.2. Experimental Models

Wild-type zebrafish (*Danio rerio* H.) were obtained from the Zebrafish International Resource Center (ZIRC, Eugene, OR, USA) and mated, staged, raised, and processed as described in the zebrafish handbook [36]. Fertilized zebrafish eggs were obtained from naturally spawned wild type and transgenic fish maintained in our facilities following standard husbandry practices. The animals were maintained on a 12 h light/dark cycle at 28 °C. Nine twelve-month-old transparent roy^a9/a9^; nacre^w2/w2^ (Casper) zebrafish, in which pigment cell production is impaired [37], were used for transplantation.

### 2.3. Tumor Generation Using MiniCoopR

Tumors were generated using the minicoopR technology, in which a plasmid allows to place the gene of interest (GOI) under the *mitfa* promoter and contains a *mitfa* minigene that leads to melanocyte rescue in Casper zebrafish, allowing easy tracking of which cells are transformed due to melanin expression [35,38]. MiniCoopR *mitfa:NRAS-Q61R* (Addgene plasmid #118847; http://n2t.net/addgene:118847; RRID:Addgene_118847) and MiniCoopR mitfa:EGFP (Addgene plasmid #118850; http://n2t.net/addgene:118850; RRID:Addgene_118850) were a gift from Leonard Zon and were previously described [39]. The MiniCoopR *mitfa:DN-DUOX1* was generated by MultiSite Gateway assemblies using LR Clonase II Plus (ThermoFisher Scientific, Madrid, Spain) according to standard protocols and using previously described Tol2kit vectors [40] and a truncated, dominant negative (DN) form of the wild-type zebrafish DUOX1, which lacks the entire flavin domain (residues 1–1232) and robustly inhibits endogenous DUOX1 [41]. Twenty-five pg of each MiniCoopR plasmid and 25 pg of Tol2 transposase mRNA were microinjected into one-cell Casper zebrafish embryos. Larvae were screened at 5 dpf to select those with melanocyte rescue. These fish were tracked and scored monthly for three months. The fish were classified into five groups according to repigmentation state and the development of tumors, as follows. (1) Normal, fish with no visible repigmentation; (2) Low repigmentation, fish with some rescued melanocytes dispersed throughout the body; (3) Nevus, fish with a local growth of these rescued melanocytes; (4) Black, fish with extensive repigmentation but no visible nodular tumors; (5) Nodular tumor, fish with clear vertical growth forming nodular tumors in the fish.

### 2.4. Tumor Sampling

Tumors from donor fish euthanized with buffered tricaine were excised with a scalpel, placed in 1 mL of dissection medium [DMEM/F12 (ThermoFisher Scientific), 100 UI/mL penicillin, 100 μg/mL streptomycin, 0.075 mg/mL Liberase (Roche, Madrid, Spain)] and the Liberase was allowed to act for at least 15 min at room temperature. The dissection medium and tumor mass were then placed in a petri dish and manually disaggregated with a clean razor blade. After this step, the cells are recovered and washed with washing medium [DMEM/F12 with penicillin/streptomycin and 15% heat-inactivated fetal bovine serum (FBS, ThermoFisher Scientific)]. The cell suspension was filtered twice through a 40 μm filter (Beckton Dickinson, Franklin Lakes, NJ, USA), counted, centrifuged at 800× *g* for 5 min at 4 °C, and resuspended in PBS containing 5% FBS to give a final concentration of 100,000 cells/uL.

### 2.5. Allotransplant in Adult Zebrafish

Adult zebrafish used as recipients were immunosuppressed to avoid rejection of the donor material. They were anesthetized using a dual anesthetic protocol to minimize overexposure to tricaine [42]. Briefly, fish were treated with tricaine and then transferred to tricaine/isoflurane solution. Afterward, fish were treated with split sublethal dose (30 Gy) of X-irradiation (YXLON SMART 200E, 200 kV, 4.5 mA) 2 days before transplantation. Around 20 anesthetized fish per tumor were injected with 300,000 cells into the dorsal subcutaneous cavity using a 26S-gauged syringe (Hamilton, Reno, NV, USA). The syringe was washed in 70% ethanol and rinsed with PBS between uses. After transplantation, fish were placed into a recovery tank with fresh fish water and returned to the system. Finally, these transplanted fish were tracked weekly for a month, anesthetized, and photographed with a mounted camera (Nikon D3100 with a Nikon AF-S Micro Lens) to follow the growth of tumor cells in the injection site. The size of the pigmented tumor was then measured using Adobe Photoshop 2022 as the number of pigmented pixels. Tumor scoring was blinded, and experiments were independently repeated at least three times. For the DN-DUOX1 condition, five different tumors were transplanted, while for the GFP condition, three different tumors were used.

### 2.6. Statistical Analysis

Data are shown as mean ± SEM. The survival curves were analyzed by log-rank (Mantel-Cox) test. Differences between the two samples were analyzed using a Student *t*-test. Contingency graphs were analyzed using the Chi-square (and Fisher’s exact) test and correlation studies with Pearson’s correlation coefficient. The sample size for each treatment is indicated in the graph and/or in the figure legend. Statistical significance was defined as * *p* < 0.05, ** *p* < 0.01, *** *p* < 0.001, **** *p* < 0.0001.

## 3. Results

### 3.1. DUOX1 Expression Is Associated with the Prognosis of Primary Melanoma Patients

The survival rate of SKCM patients of TCGA cohort stratified according to their expression of *DUOX1* was analyzed. The patient data were included in four different quartiles, being the first quartile, low expression; second and third quartile, medium expression; and the fourth quartile, high expression. Surprisingly, high *DUOX1* transcript levels in early-stage SKCM were associated with poor patient survival, while low *DUOX1* levels showed a similar prognosis to the patient with medium levels (Figure 1A). However, no statistically significant differences in patient survival were observed associated with *DUOX1* transcript levels in late-stage SKCM (Figure 1A). As expected, the survival of late-stage SKCM patients was significantly lower than that of early-stage SKCM patients (Figure 1B). Furthermore, *DUOX1* mRNA expression was significantly lower in metastatic than in primary SKCM (Figure 1C).

### 3.2. Melanocyte DUOX1 Inhibition Does Not Affect Melanocyte Transformation and Early SKCM Progression

We used a genetic strategy based on the ability of a DN form of zebrafish DUOX1, lacking the entire flavin domain, to robustly inhibit endogenous DUOX1 [41]. MinicoopR mitfa:DN-DUOX1 and MinicoopR mitfa:NRAS-Q61R constructs were microinjected into one-cell Casper zebrafish embryos to express oncogenic NRAS and inhibit DUOX1 in melanocytes. As a control, eGFP was injected instead of DN-DUOX1. Fish were examined at 5 dpf to select those with more than 10 rescued melanocytes, and then fish were followed for three months, and melanomas were scored (Figure 2A). Melanomas were categorized into five groups depending on their repigmentation or appearance: I normal, II low repigmentation, III nevus, IV black, and V nodular tumor (Figure 2B). The first group included fish with no melanocyte rescue, the second group included fish with some isolated melanocytes, the third group consisted of fish with a larger area of melanocytes growing together to form a nevus, the fourth group included fish with more than 50% of the body showing melanocyte rescue but no presence of tumor growing on the skin, and the fifth group included fish with the presence of nodular tumor usually on the dorsal, anal or caudal fins, or on the back of the fish. Although the difference between nodular and black groups was established when vertical growth could be seen, the fourth group could contain some fish with nodular tumors.

While at 30 dpf, the fish injected with MinicoopR *mitfa:NRAS-Q61R* and *mitfa:DN-DUOX1* showed a higher percentage of nodular tumors at 30 dpf, this effect was no longer observed at later time points and showed no clear and statistically significant differences (Figure 2C,D).

### 3.3. DUOX1 Inhibition Autonomously Reduces Aggressiveness and Growth of Transplanted Melanomas

To investigate the role of DUOX-1 in melanoma aggressiveness and invasiveness, 300,000 melanoma cells developed with the minicoopR system were transplanted into the dorsal sinus of one-year old adult Casper zebrafish recipients previously irradiated with 30 Gy. SKCM development was monitored weekly for one month. Both sides of the fish were imaged to visualize tumor cell growth and dissemination in vivo over time (Figure 3A).

Tumor engraftment was visible at 7 days post-transplantation in both conditions (Figure 3B), and 41.7% control and 46.0 DN-DUOX1 were still alive at the end of the experiments. The recipient fish transplanted with DUOX1-deficient SKCMs showed significantly smaller tumors than their control counterparts from the first week after transplantation until the end of the experiment (Figure 3C). In addition, recipients transplanted with DUOX1-deficient SKCMs showed a significantly lower growth rate than fish transplanted with control SKCMs (Figure 3D). The sex of the recipient fish did not affect the growth of either control or DUOX-1-deficient SKCMs (data not shown). Interestingly, fish transplanted with DUOX1-deficient SKCMs showed a higher incidence of cells migrating to distant parts of the injection area (metastasis) than those transplanted with control SKCMs (Figure 4).

## 4. Discussion

SKCM is the fifth most common cancer worldwide and the most aggressive skin cancer due to its high metastatic capacity. Increasing evidence suggests that oxidative stress is involved in the development of several chronic diseases and in the transformation and progression of many common cancers, including SKCM. Complex crosstalk between oxidative stress and inflammatory and immune responses has been shown [43]. The inflammatory response induces the recruitment of innate and adaptive immune cells within the tumor, which, in turn, can induce oxidative stress that is involved in many steps of SKCM development, such as DNA damage and mutation of SKCM-associated genes, cell metabolism, response to hypoxia, tumor immunity and metastasis [44].

It is well established that ROS plays a crucial role in the early development, progression, and suppression of many types of cancer, including SKCM [33]. In this study, we have focused on DUOX1, an important NADPH oxidase highly expressed in the skin [45]. We found that high DUOX1 transcript levels robustly correlated with a poor prognosis of patients with early-stage, but no late-stage, SKCM. These results suggest the importance of oxidative stress balance in melanoma progression and the dramatic impact of DUOX1-derived ROS dysregulation on the prognosis of SKCM patients. This observation can be explained by considering the dual role of ROS in cancer [46]. Thus, it has been shown that, on the one hand, elevated ROS production could cause DNA damage and lead to oncogene activation or anti-oncogene inactivation, which facilitates tumor progression. Under these conditions, cancer cells require increased energy levels to maintain abnormal growth rates and excessive metabolism, resulting in increased ROS production [46]. On the other hand, cancer cells may attract immune cells and promote an inflammatory microenvironment that favors tumor cell proliferation [47,48]. This may explain why antioxidants suppress cancer initiation in some settings [49,50] while increasing it in other settings [51,52]. Thus, while several articles have shown that increasing dietary antioxidants does not reduce cancer incidence [53,54,55,56,57], others show the good properties of some dietary flavonoids [58] and polyphenols [59], as antioxidants by suppressing cancer via tumor protein P53 (TP53) signaling pathway. Therefore, despite the commercial popularity of a huge amount of antioxidant molecules and their benefits, the role of antioxidants in therapy for many diseases, including cancer, needs to be determined and requires more investigation to shed light on their complex effects [60,61,62].

The impact of ROS in SKCM appears to be even more complex, as SKCM cells in the blood experienced oxidative stress that was not observed in subcutaneous tumors. Thus, oxidative stress limits the distant metastasis of SKCM cells in vivo [23]. These results are consistent with the poor survival of early-stage SKCM patients with high expression of *DUOX1* and with the downregulation of *DUOX1* in metastatic SKCM. Furthermore, they suggest that DUOX1 plays a dual role in SKCM, as confirmed by our zebrafish model with inhibition of DUOX1 in SKCM cells. Although we found no significant effect of this enzyme on melanocyte transformation and early SKCM progression, allotransplantation experiments in pre-irradiated, i.e., immunocompromised, zebrafish recipients revealed that DUOX1-deficient SKCM cells showed reduced growth but, paradoxically, generated more metastasis than control SKCMs. Therefore, DUOX1-derived ROS may facilitate primary SKCM growth but limit distant metastasis, as has been shown with xenotransplanted human SKCM cell lines in immunocompromised mice [23]. Surprisingly, our results revealed that inhibition of DUOX1 enzymatic activity restrained the growth of transplanted SKCMs in immunocompromised recipients but not in fully immunocompetent individuals, suggesting that the immune response may counteract the enhanced metastatic potential of DUOX1-deficient SKCM cells in zebrafish. Future experiments overexpressing DUOX1 in SKCM cells and using immunocompromised Casper zebrafish individuals injected with MinicoopR would be very informative in clarifying these questions.

## 5. Conclusions

We have found that high levels of *DUOX1* expression were associated with the poor survival of early-stage SKCM patients but not with the prognosis of late-stage SKCM patients. In addition, *DUOX1* transcript levels declined in metastatic SKCM. In addition, zebrafish models of SKCM revealed that DUOX-1-deficient SKCMs showed reduced tumor growth, but increased metastasis compared to control SKCMs. As this effect is only observed in immunocompromised individuals, the immune system appears to be able to counteract this elevated metastatic potential of DUOX1-deficient SKCMs. Collectively, our results highlight the complex crosstalk between oxidative stress and immunity in cancer and point to the relevance of DUOX1 in SKCM.

## Figures and Tables

**Figure 1 antioxidants-12-00708-f001:**
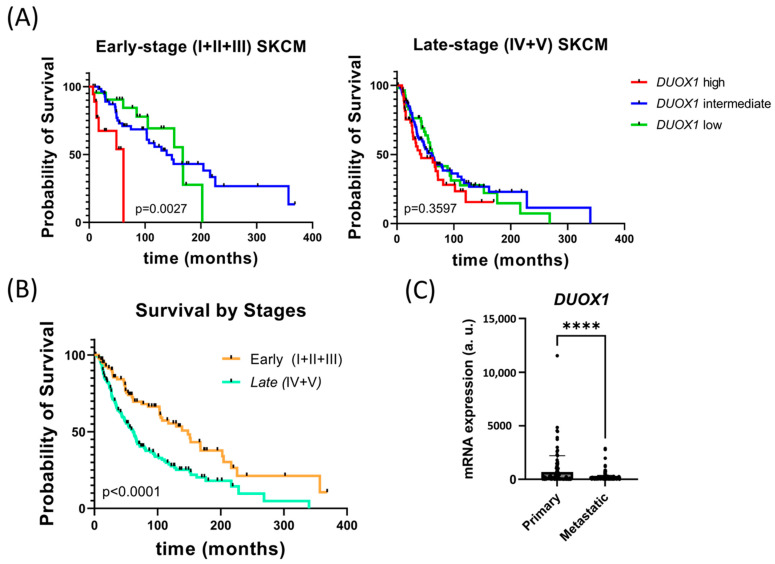
***DUOX1* expression correlates with survival in SKCM patients and decreases in metastatic SKCM.** (**A**) Kaplan–Meier survival analysis of TCGA cohort of early- (I + II + III Clark levels at diagnosis) and late- (IV and V Clark levels at diagnosis) stage SKCM patients according to their *DUOX1* transcript levels: first quartile (Low), second and third (Medium) and fourth (High). Early-stage melanoma: Low, *n* = 22; Medium, *n* = 59; High, *n* = 16. Late-stage melanoma: Low, *n* = 57; Medium, *n* = 100; High, *n* = 60. Statistical significance was calculated with Mantel–Cox test. (**B**) Kaplan–Meier survival analysis of TCGA cohort of SKCM patients according to Clark-level classification at diagnosis: early-stage (I + II + III, *n* = 99) and late-stage (IV + V, *n* = 217). Statistical significance was calculated with Mantel–Cox test. (**C**) DUOX1 transcript levels in primary (*n* = 102) and metastatic (*n* = 360) SKCMs from TCGA cohort. Each dot represents a patient, and the mean is also shown. **** *p* < 0.0001 according to unpaired Student *t* test. a.u., arbitrary units.

**Figure 2 antioxidants-12-00708-f002:**
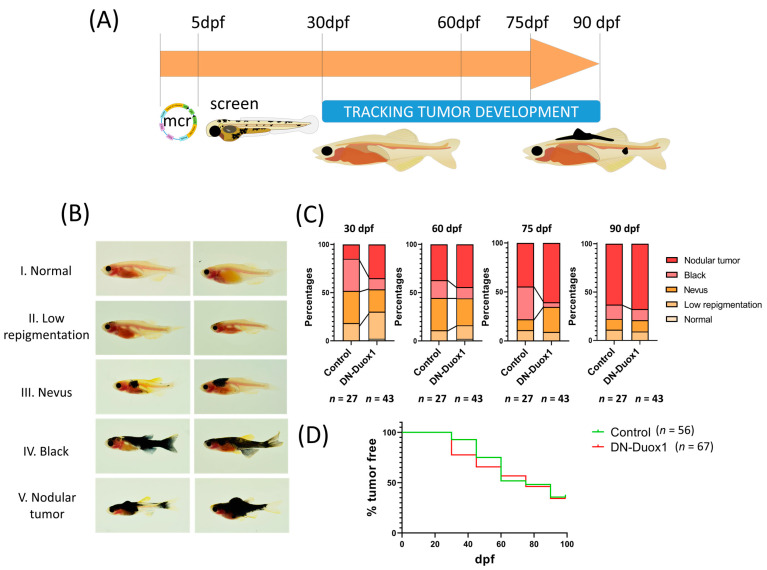
**Melanocyte DUOX1 inhibition does not affect melanocyte transformation and early melanoma progression.** (**A**) Schematic representation of the procedure to co-express oncogenic NRAS-Q61R and DN-DUOX1 in melanocytes. Zebrafish one-cell Casper zebrafish embryos were injected with MinicoopR *mitfa:NRAS-Q61R* and either MinicoopR *mitfa:DN-DUOX1* or MinicoopR *mitfa:EGFP* (control). Larvae were examined at 5 dpf for the presence of melanocytes and images were acquired monthly for 3 months to track melanoma development. (**B**) Representative images of the five different categories established to classify tumor progression. (**C**) Percentages of fish in the different categories at 30, 60, 75 and 90 dpf. (**D**) Tumor free curve. Representation of the percentage of fish without nodular tumors.

**Figure 3 antioxidants-12-00708-f003:**
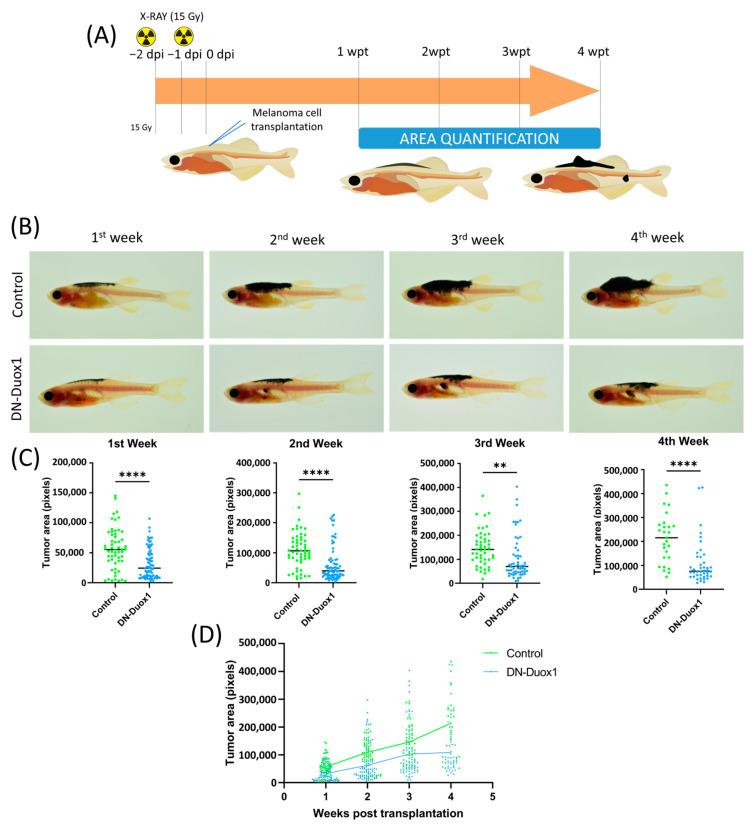
**DUOX1 inhibition autonomously reduces aggressiveness and growth of transplanted SKCMs.** (**A**) Schematic diagram showing adult allotransplantation procedure. One-year-old Casper zebrafish were irradiated 2 days before transplantation to prevent tumor rejection. Three hundred thousand cells from the nodular tumors of either MinicoopR *mitfa:NRAS-QW61R*/*mitfa:DN-DUOX1* or MinicoopR *mitfa:NRAS-Q61R*/*mitfa:EGFP* (control) fish were subcutaneously injected in pre-irradiated recipients, and images were taken weekly during the following 4 weeks after transplantation and analyzed as indicated in the Methods and Materials section. Arrow, timeline; dpi, days post-injection; wpt, weeks post-transplant. (**B**) Representative images of transplanted melanoma growth rate of MinicoopR *mitfa:NRAS-QW61R*/*mitfa:DN-DUOX1* and MinicoopR *mitfa:NRAS-Q61R*/*mitfa:EGFP* in pre-irradiated adult Casper zebrafish. (**C**) Average tumor size for each week post-transplant. Each dot represents a recipient-transplanted fish, and the mean is also shown. ** *p* < 0.01, **** *p* < 0.0001 according to unpaired Student *t* test. (**D**) Growth rate of transplanted MinicoopR *mitfa:NRAS-QW61R*/*mitfa:DN-DUOX1* and MinicoopR *mitfa:NRAS-Q61R*/*mitfa:EGFP* SKCMs. DN-DUOX1: *n* = 5 tumors and 111 recipient fish; EGFP: *n* = 3 tumors and 72 recipient fish.

**Figure 4 antioxidants-12-00708-f004:**
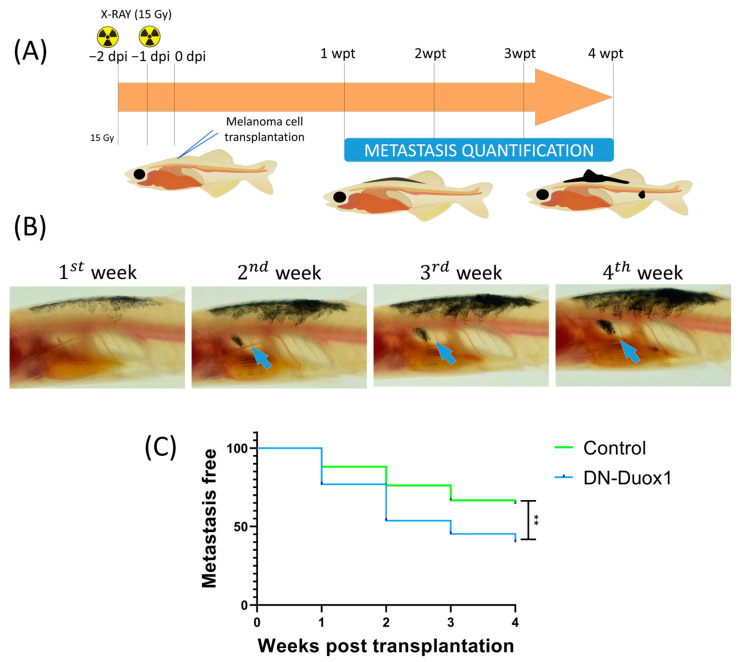
**DUOX1 deficiency in SKCM promotes metastasis.** (**A**) Schematic diagram of adult allotransplantation assays. (**B**) Representative images of the progression of metastasis (arrows). (**C**) Metastasis-free curve of adult zebrafish transplanted with MinicoopR *mitfa:NRAS-Q61R*/*mitfa:DN-DUOX1* and MinicoopR *mitfa:NRAS-Q61R*/*mitfa:EGFP* (control). ** *p* < 0.01 according to a Log rank Mantel–Cox test. DN-DUOX1: *n* = 5 tumors and 111 recipient fish; EGFP: *n* = 3 tumors and 72 recipient fish.

## Data Availability

Data is contained within the article.

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
