# Peer review of "Dual Role of DUOX1-Derived Reactive Oxygen Species in Melanoma"

_antioxidants, 2023, doi:10.3390/antiox12030708_

Round 1
Reviewer 1 Report
This manuscript is a study on the caner regulation of dual oxidase1. A model using Zebrafish was well produced, inhibiting of cell proliferation by dual oxidase1 was well demonstrated and appropriate results were obtained for the effects to melanoma. The author reported the possibility that dual oxidase1 can inhibit cancer growth. So, the author showed the potential for new therapy pathway through by dual oxidase1 in the future.
The author has already provided sufficient review on the related research. (Zebrafish Models to Study the Crosstalk between Inflammation and NADPH Oxidase-Derived Oxidative Stress in Melanoma. Antioxidants (Basel). 2022 Jul; 11(7): 1277.)
Therefore, it will be an excellent paper if the following results are revised and supplemented.
- The author mentioned mostly melanoma in introduction. Rather, it is more appropriate to change the content to cancer research using double oxidase and zebrafish.
- The dual oxidase 1 may be associated with the immune system. Therefore, mRNA analysis for cytokines (interlukin, INF.. etc) and apoptosis is necessary.
- Analysis of expected pathway of double oxidase1 to cancer inhibition. (Ex. MAPK, Nkx2. EGFR, etc)

Author Response
This manuscript is a study on the caner regulation of dual oxidase1. A model using Zebrafish was well produced, inhibiting of cell proliferation by dual oxidase1 was well demonstrated and appropriate results were obtained for the effects to melanoma. The author reported the possibility that dual oxidase1 can inhibit cancer growth. So, the author showed the potential for new therapy pathway through by dual oxidase1 in the future.
The author has already provided sufficient review on the related research. (Zebrafish Models to Study the Crosstalk between Inflammation and NADPH Oxidase-Derived Oxidative Stress in Melanoma. Antioxidants (Basel). 2022 Jul; 11(7): 1277.)
Therefore, it will be an excellent paper if the following results are revised and supplemented.
We thank these comments on our paper.
- The author mentioned mostly melanoma in introduction. Rather, it is more appropriate to change the content to cancer research using double oxidase and zebrafish.
We have explained in the introduction of the revised version the evidences supporting a key role of DUOX1 in cancer initiation, progression, and resistance to therapy. We have also introduced the relevance of the zebrafish cancer model.
- The dual oxidase 1 may be associated with the immune system. Therefore, mRNA analysis for cytokines (interlukin, INF.. etc) and apoptosis is necessary.
- Analysis of expected pathway of double oxidase1 to cancer inhibition. (Ex. MAPK, Nkx2. EGFR, etc)
We thank these suggestions. The relevance of the immune system and the signaling mechanism involved in the impact of Duox1 in melanoma is very complex and required transcriptomic, in vivo imaging and functional studies. This is out of the scope of the present manuscript and will be analyzed in future studies.
Reviewer 2 Report
The aim of the study described in this paper was to investigate the role of dual oxidase 1 (DUOX1) in melanoma aggressiveness in in vivo model. The study was welldesigned and documented, and may be considered of potential interest in the area of melanoma research. Before publication a few changes and additions should be enter into the manuscript. Specific comments are below.
Abstract
Abstract in my opinion needs rewriting. It is true that the journal does not require it to have a structured form, but elements such as the background, the aim, a description of the research methodology, a description of the obtained result, possibly also conclusions, should be included in it. In its current form, it contains only the background, a description of the results and the final conclusion.
Introduction & Discusion
I suggest that the information on the position of melanoma in the ranking of the most common cancers and other statistics data should be based on global data, e.g. Sung et al 2021 CA CANCER J CLIN 2021, 71, 209-249. doi: 10.3322/caac.21660.
Results
There is no information which color in Fig. 2C has been assigned to the selected category (I-V) of Fig. 2B.
Author Response
The aim of the study described in this paper was to investigate the role of dual oxidase 1 (DUOX1) in melanoma aggressiveness in in vivo model. The study was welldesigned and documented, and may be considered of potential interest in the area of melanoma research. Before publication a few changes and additions should be enter into the manuscript. Specific comments are below.
Abstract
Abstract in my opinion needs rewriting. It is true that the journal does not require it to have a structured form, but elements such as the background, the aim, a description of the research methodology, a description of the obtained result, possibly also conclusions, should be included in it. In its current form, it contains only the background, a description of the results and the final conclusion.
The abstract has been rewritten to include our aim and a description of the methodology used.
Introduction & Discusion
I suggest that the information on the position of melanoma in the ranking of the most common cancers and other statistics data should be based on global data, e.g. Sung et al 2021 CA CANCER J CLIN 2021, 71, 209-249. doi: 10.3322/caac.21660.
This citation was included in the revised version.
Results
There is no information which color in Fig. 2C has been assigned to the selected category (I-V) of Fig. 2B.
We are sorry for this mistake. The panel has been added to Fig. 2C in the revised version.
Reviewer 3 Report
In this study Dr Pardo-Sànchez and co-workers start from the observation that primary melanoma patients with high levels of Dual Oxidase 1 (DUOX1) gene expression have a shorter life expectancy than those with low DUOX1 expression. This difference is not observed in metastatic melanoma in which the survival expectancy is independent from DUOX1 levels of expression. In general, however, primary melanoma express higher level than DUOX1 than metastatic melanoma. Then the authors inhibit Duox1 generating transgenic zebrafish and with a dominant negative construction, but do not observe melanocyte transformation and early melanoma progression. Finally they observe that inhibition of Duox1 reduces the aggressiveness and growth of melanomas transplanted in zebrafish. They also show that reduced level of Duox1 are associated to higher metastatic potential.
The findings presented are interesting, but somehow confusing and the relation between the “human” part of the paper and the “zebrafish” part is not convincing.
I have several points that need to be addressed before considering the paper for publication:
1. The introduction needs to elaborate much better on the potential use of Zebrafish as a model for melanoma. The authors have just publishes a review on this topic on this same journal, they only cite the review in the discussion, but it is important to take the time to discuss different aspects of Zebrafish applications to melanoma. It is essential to explain the relation human/fish models and elaborate on this point.
2. The statistics of Figure 1A is not cleat to me. The p values correspond only to the comparison High/Low? What about the intermediate level?
The number of patients in each category is not shown, the total number of patientd is shown only in the methods section not in legend ot text.
This analysis is in my view quite superficial. One point I do not understand is that, at first view, comparing figure 1A right and left, the survival of metastatic melanoma is higher than that of primary melanoma….
Studies from « The Central Malignant Melanoma Registry (CMMR) in Germany show a 10 years survival (240 month) of primary melanomas between 98 and 85% depending on the stage at diagnosis, while, several studies on malignant metastatic melanoma show a much shorter life expectancy. The authors need to explain, if not I would question the origin of the data they analyze
3. In my view the correlations with the “reactome” are purely speculative.
4. In figure 2 the authors claim to have categorized their animals in five groups, but in Fig. 2C only 4 groups are shown and we do not know the colour code. In the figure I do not see any “ns” to indicate non significant.
5. Are the authors sure that all the fish analyzed in Fig.2 have a reduced level of Duox1 expression, this expression can be quite variable particularly with a Dominant Negative. They should have checked after the experiment and see if the level of Duox1 expression did correlate with the pigmentation. It seems important.
6. Regarding the metastatic potential in fish only one animal is shown at different ages, no numbers are shown, how many animals did you look at? How many had metastasis? Only % are shown. In any case about 30% of fish non transplanted with DN-Duox show metastasis.
7. In the conclusion I am very confused by the sentence : “We have found that both high and low levels of DUOX1 expression are associated with poor survival of melanoma patients.” Checking Fig.1 it seems that high level are associated with poor survival, lower levels with metastatic potential, but, as mentioned above in Fig. 1 Metastatic Melanoma survive as much or more than primary melanoma. I do not understand the conclusion (and the title….). You need to express much better your thought.
Minor point:
I would specify “Casper Zebrafish” instead of “Casper” in all the paper and “one-cell embryos” instead of “one-cell eggs” in section 3.2
Author Response
In this study Dr Pardo-Sànchez and co-workers start from the observation that primary melanoma patients with high levels of Dual Oxidase 1 (DUOX1) gene expression have a shorter life expectancy than those with low DUOX1 expression. This difference is not observed in metastatic melanoma in which the survival expectancy is independent from DUOX1 levels of expression. In general, however, primary melanoma express higher level than DUOX1 than metastatic melanoma. Then the authors inhibit Duox1 generating transgenic zebrafish and with a dominant negative construction, but do not observe melanocyte transformation and early melanoma progression. Finally they observe that inhibition of Duox1 reduces the aggressiveness and growth of melanomas transplanted in zebrafish. They also show that reduced level of Duox1 are associated to higher metastatic potential.
The findings presented are interesting, but somehow confusing and the relation between the “human” part of the paper and the “zebrafish” part is not convincing.
I have several points that need to be addressed before considering the paper for publication:
- The introduction needs to elaborate much better on the potential use of Zebrafish as a model for melanoma. The authors have just publishes a review on this topic on this same journal, they only cite the review in the discussion, but it is important to take the time to discuss different aspects of Zebrafish applications to melanoma. It is essential to explain the relation human/fish models and elaborate on this point.
We have included a novel paragraph describing the zebrafish models of melanoma.
- The statistics of Figure 1A is not cleat to me. The p values correspond only to the comparison High/Low? What about the intermediate level?
The number of patients in each category is not shown, the total number of patientd is shown only in the methods section not in legend ot text.
We have indicated the number of patients in each group and the statistical analysis performed in the legend to Fig. 1.
This analysis is in my view quite superficial. One point I do not understand is that, at first view, comparing figure 1A right and left, the survival of metastatic melanoma is higher than that of primary melanoma….
Studies from « The Central Malignant Melanoma Registry (CMMR) in Germany show a 10 years survival (240 month) of primary melanomas between 98 and 85% depending on the stage at diagnosis, while, several studies on malignant metastatic melanoma show a much shorter life expectancy. The authors need to explain, if not I would question the origin of the data they analyze
You are right. The classification primary and metastatic refers to the biopsy. Therefore, we have used the Clark levels at diagnosis and stratified the patients as early-stage (I+II+III) and late-stage melanoma (IV+V).
- In my view the correlations with the “reactome” are purely speculative.
We have deleted these data.
- In figure 2 the authors claim to have categorized their animals in five groups, but in Fig. 2C only 4 groups are shown and we do not know the colour code. In the figure I do not see any “ns” to indicate non significant.
We are sorry for this mistake. We have added the panel to Fig. 2C that correspond to the categories shown in Fig. 2B. The legend was corrected and non-significant was deleted.
- Are the authors sure that all the fish analyzed in Fig.2 have a reduced level of Duox1 expression, this expression can be quite variable particularly with a Dominant Negative. They should have checked after the experiment and see if the level of Duox1expression did correlate with the pigmentation. It seems important.
The dominant negative does not affect expression but rather inhibit endogenous Duox1. This is mentioned in the M&M section with an appropriate citation (28). We have now added this information in the Result section to facilitate interpretation of the results to the readers.
- Regarding the metastatic potential in fish only one animal is shown at different ages, no numbers are shown, how many animals did you look at? How many had metastasis? Only % are shown. In any case about 30% of fish non transplanted with DN-Duox show metastasis.
This information was indicated in the legend to Figure 3 and has also now include in the legend to Figure 4. We analysed 111 recipient fish for DN-Duox1 tumors 72 recipient fish for EGFP tumors.
- In the conclusion I am very confused by the sentence : “We have found that both high and low levels of DUOX1 expression are associated with poor survival of melanoma patients.” Checking Fig.1 it seems that high level are associated with poor survival, lower levels with metastatic potential, but, as mentioned above in Fig. 1 Metastatic Melanoma survive as much or more than primary melanoma. I do not understand the conclusion (and the title….). You need to express much better your thought.
This has been rewriting according to the novel analysis that we performed based using the Clark classification at diagnosis. Thanks for this valuable suggestion.
Minor point:
I would specify “Casper Zebrafish” instead of “Casper” in all the paper and “one-cell embryos” instead of “one-cell eggs” in section 3.2
Thanks for these suggestions. We have corrected them in the revised version.
Reviewer 4 Report
The paper is interesting and of some new added value, however in my opinion it demands supplementation, and this is the introductory condition to accept the paper for publication
1. The character of the experiments described in the paper is generally descritive, the results of the mechanisms are obtained mainly indirectly, however the actual mechanisms of the inhibition of growth and acceleration of metastasizing should be additionally supported directly, by checking and describing mechanisms. In particular, the level of oxidative free radical species and the expression of immunological modulators of the effects should be directly shown. In melanoma, the activation/inhibition of the tumor via influencing immunity seems crucial, and it should be proven rather than speculated. This panel of experiments should supplement the present, careful but limited observation of melanoma growth. In particular, the conclusions concerning the humat effects are supported only partially, as documented by the observations of the Danio rerio model.
2. The ethical aspect of the experiments with the application of Danio rerio should be documented. The lack of the formal bioethical opinion should be supplemented always, and the permission obtained if the Danio fish is older that 5 days after spawn.
3. The statistics anova study are performed, by the authors support it by the Student;s t-test, which would be justified only in the normal distribution of the data (and this should be tested independently, if not , a non-parametric test should be carried on. Similarly, what is the reason to evaluate the variance using the SEM instead of the proper evaluator - Standard deviation of the means (SD). SE evaluates mainly the difference between the actual value and the estimated value obtained by averaging particular (finite) number of experimental measurements. SD posesses a real iinterpretation of the population of samples, indicating the actual dispersion of the values. At least somme additional explanations are here expected.
Author Response
The paper is interesting and of some new added value, however in my opinion it demands supplementation, and this is the introductory condition to accept the paper for publication
- The character of the experiments described in the paper is generally descritive, the results of the mechanisms are obtained mainly indirectly, however the actual mechanisms of the inhibition of growth and acceleration of metastasizing should be additionally supported directly, by checking and describing mechanisms. In particular, the level of oxidative free radical species and the expression of immunological modulators of the effects should be directly shown. In melanoma, the activation/inhibition of the tumor via influencing immunity seems crucial, and it should be proven rather than speculated. This panel of experiments should supplement the present, careful but limited observation of melanoma growth. In particular, the conclusions concerning the humat effects are supported only partially, as documented by the observations of the Danio rerio model.
We thank these suggestions. The relevance of the immune system and the signaling mechanism involved in the impact of Duox1 in melanoma is very complex and required transcriptomic, in vivo imaging and functional studies. This is out of the scope of the present manuscript and will be analyzed in future studies.
- The ethical aspect of the experiments with the application of Danio rerio should be documented. The lack of the formal bioethical opinion should be supplemented always, and the permission obtained if the Danio fish is older that 5 days after spawn.
This was already indicated at the end of the manuscript in the Institutional Review Board Statement, following the format of MDPI journals.
- The statistics anova study are performed, by the authors support it by the Student;s t-test, which would be justified only in the normal distribution of the data (and this should be tested independently, if not , a non-parametric test should be carried on. Similarly, what is the reason to evaluate the variance using the SEM instead of the proper evaluator - Standard deviation of the means (SD). SE evaluates mainly the difference between the actual value and the estimated value obtained by averaging particular (finite) number of experimental measurements. SD posesses a real iinterpretation of the population of samples, indicating the actual dispersion of the values. At least somme additional explanations are here expected.
We did not use ANOVA. This was a mistake and has been removed in the revised version. As indicated in section 2.6, we showed the results as mean ± SEM but we did not use SEM, but SD, for statistical analysis.
Reviewer 5 Report
The study demonstrates dual role of DUOX1-derived reactive oxygen species in melanoma. The dual role of ROS is very important to be investigated.
In Results, it is described that the first quartile was low expression of DUOX1. More detailed explanation for the differences between the four different quartiles with regards to Figure 1 in Results section.
It seems that MiniCoopR is a main technique in this study. The MiniCoopR may be explained briefly atain to remind the readers in the last sentence of Discussion.
Author Response
The study demonstrates dual role of DUOX1-derived reactive oxygen species in melanoma. The dual role of ROS is very important to be investigated.
In Results, it is described that the first quartile was low expression of DUOX1. More detailed explanation for the differences between the four different quartiles with regards to Figure 1 in Results section.
We have changed Figure 1 following the concerns of reviewer 3 and have now added more information to its legend. In addition, we have indicated that the stratification based in DUOX1 expression was done using the quartiles.
It seems that MiniCoopR is a main technique in this study. The MiniCoopR may be explained briefly atain to remind the readers in the last sentence of Discussion.
Thanks for this suggestion. We have now explained the MiniCoopR technology in the Introduction.
Round 2
Reviewer 3 Report
The authors have replied well to all my questions and the paper can now be published. I think it is still needed a re-reading to correct minor english errors.
Author Response
The authors have replied well to all my questions and the paper can now be published. I think it is still needed a re-reading to correct minor english errors.
We have carefully read the manuscript and corrected several typos and mistakes.
Reviewer 4 Report
The manuscript has been extensively improved, according to my opinions in the preview review, it has reached the level to be accepted for publication. In my opinion, now the manuscript is ready to be published.
Author Response
The manuscript has been extensively improved, according to my opinions in the preview review, it has reached the level to be accepted for publication. In my opinion, now the manuscript is ready to be published.
We thanks the constructive comments that helped us to improved the manuscript.
